# Application of UAV Digital Photogrammetry in Geological Investigation and Stability Evaluation of High-Steep Mine Rock Slope

Jianning Hao [1,2], Xiuli Zhang [1,*], Chengtang Wang [1], Hao Wang [1] and Haibin Wang [1]

[1] State Key Laboratory of Geomechanics and Geotechnical Engineering, Institute of Rock and Soil Mechanics, Chinese Academy of Sciences, Wuhan 430071, China; haojianning20@mails.ucas.ac.cn (J.H.); ctwang@whrsm.ac.cn (C.W.); hwang@whrsm.ac.cn (H.W.); hbwang@whrsm.ac.cn (H.W.)

[2] University of Chinese Academy of Sciences, Beijing 100049, China

\* Correspondence: zxl@whrsm.ac.cn

**Abstract:** For the stability analysis of rock slope, it is very critical to obtain the spatial geometric characteristics of the structural surfaces of the rock mass accurately and effectively. As for a high-steep rock slope of an iron ore mine, in order to solve the problems of inefficiency and high risk of traditional manual geological survey, the geological survey and stability evaluation of the slope was carried out by adopting unmanned aerial vehicle digital photogrammetry (UAV-DP) technology. Firstly, a large number of high-resolution images of the slope were obtained by UAV-DP. Then, the structure from motion (SFM) method was used to construct the fine 3D point cloud model of the slope, which was subjected to coplanarity detection and *K*-means clustering for identifying the structural surfaces. Finally, the stability and failure model of the slope cut by the structural surfaces are analyzed by using the stereo-projection and discrete element methods. The research results show that the error between UAV-DP and manual measurement is within the acceptable range, which demonstrates the reliability of UAV-DP used in the geological investigation. Furthermore, the stability state and failure model of the slope is also consistent well with the field observation.

**Keywords:** rock slope; unmanned aerial vehicle digital photogrammetry (UAV-DP); discontinuity identification; stability evaluation

## 1. Introduction

The spatial distribution condition and relationship of the rock structure surface are pivotal to the stability of rock slopes [1–3]. In 1978, the International Rock Mechanics Society (IRSM) [4] proposed a set of recommended parameters for describing rock structural surfaces, including orientation, spacing, trace length, roughness, aperture, wall strength, filling, seepage, and the number of sets and block size. Rapid and accurate acquisition of these parameters is a prerequisite for carrying out mechanical analysis of rock masses and an important element of the geological investigation of slope sites. Traditional manual measurements rely on the professional level of investigators [5,6], with subjective bias [7] and high labor intensity and low efficiency, while the inaccessibility of steep and narrow areas also makes it difficult to carry out structural surface investigations. In view of the limitations of traditional manual survey methods, new geological survey tools and techniques have been rapidly developed and applied in recent years, such as 3D laser scanning technology and digital close-up photogrammetry. Xiu D. et al. [8] were the first in China to apply 3D laser scanning technology to the geological investigation of high-steep slopes and proposed a set of investigation and geological cataloging methods combined with engineering cases, and Bing H. [9] and Yun G. [10] conducted research on the identification and extraction methods of structural surfaces based on 3D point clouds. Based on digital

close-up photogrammetry, Feng W. [11] relied on the JX-4A digital photogrammetry workstation to quickly obtain the trace map of fissures; Dong, H. [12] applied the Lensphoto multi-baseline digital close-up photogrammetry system to establish a 3D rock model as a way to obtain structural face traces and orientation information; Bo H. [13] relied on engineering cases for the acquisition of structural faces and 3D geometric characterization method of the fracture network.

The widespread use of 3D laser scanning technology and digital close-up photogrammetry has made the acquisition of rock structure surface information fast, accurate, and efficient, but both are not easy to find suitable shooting points in complex field environments, and factors such as shooting distance and angle also tend to distort the generated models [14,15]. In contrast, UAVs are not restricted by terrain and can take multi-angle aerial photographs on near-vertical rock walls to obtain as complete an image of the study area as possible within the planned route. UAV-DP not only reduces the measurement risk and time cost but also provides rich data information for the fine reconstruction of 3D geological models. Shu J. [14] and Cheng Z. [16] used the Patch Based Multi-View Stereo (PMVS) algorithm and Structure From Motion (SFM) algorithm to model the surface images of slopes taken by UAVs in 3D and realized the application of UAV-DP in the geological investigation of high-steep slopes. Zhen Y. et al. [15] obtained topographic slope information through UAV-DP, used Hoff normal algorithm and HSV algorithm for visualizing 3D slope model reconstruction, and completed structural surface information extraction using K-mean clustering. Kong et al. [7] used a machine algorithm to automatically identify rock discontinuities based on 3D point clouds generated by UAV-DP. Shu, W. et al. [17] used the SFM algorithm to process the images from UAV-DP, identified the rock structure surface by the RANSAC algorithm, and developed a GeoSMA-3D program to analyze the slope stability. Devoto S. et al. [18] studied slow-moving coastal landslides using the UAV-DP technique and demonstrated that UAVs can acquire higher-resolution images and have more advantages in identifying discontinuity orientations and persistence. Francioni M. [19], Menegoni N. [20], and Yao H. [21] all consider that the UAV-DP technology can overcome problems related to security, elevation, steepness, and complex slope geometry. Afiqah Ismail et al. [22] investigated the combination of UAV and ground-based laser scanner for slope structural surface acquisition, and the case study showed that the combination of the two methods could obtain a complete image of the slope from the foot to the top of the slope and generate a better quality point cloud model.

In summary, the successful use of UAV-DP technology in the geological investigation of high-steep slopes shows the accuracy and reliability of UAV-DP results. It is easier to obtain a complete and clear image of the rock surface of the slope by carrying a high-resolution sensor on the UAV and extracting the information of the rock structure characteristics by using point cloud processing technology, which can provide a geological basis for slope stability evaluation. In this study, the UAV-DP technology is applied to the high-steep rock slope project on the southwest side of a certain iron ore mine, and the UAV-DP images are reconstructed on the surface to establish the point cloud model of the rock surface of the slope, and the structural surface of the rock mass is automatically identified through coplanarity detection and *K*-mean clustering, and the stability and failure characteristics of the slope are analyzed by using discrete element method based on the results of the structural surface identification.

## 2. UAV-DP Work Procedures

UAV-DP technology for slope geological survey is generally divided into the following steps (see Figure 1): (1) site survey; camera parameter calibration; (2) UAV flight route planning; (3) flight height, speed, overlap rate setting; (4) layout of ground control points (GCPs); (5) aerial photography slope image collection; (6) image processing; 3D point cloud modeling; (7) discontinuity extraction; (8) precision verification.

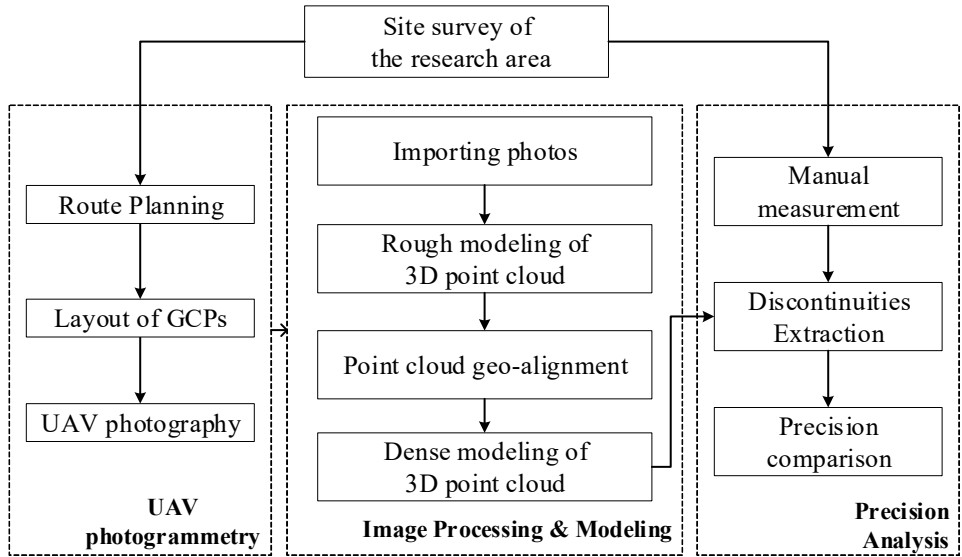

**Figure 1.** Technical flowchart.

## 2.1. UAV Type

In this study, DJI Matrice 300 RTK (M300 RTK) UAV (Figure 2) is selected. Following the weighting classification methodology of UAVs by Brooke–Holland [23], M300 RTK UAV is a mini drone. It integrates an advanced flight control system, six-way binocular vision, infrared sensing system, and 1.3-megapixel FPV camera, compatible with omnidirectional obstacle avoidance radar and equipped with six-way positioning and obstacle avoidance, and its main technical parameters are shown in Table 1. It is equipped with a 45-megapixel Zenmuse P1 full-frame image sensor on the UAV, which can ensure the acquisition of high-precision and high-resolution images. The mission of the UAV is mainly to take a set of photos with a sufficient overlap rate to capture detailed rock surface features for reconstructing a fine 3D rock surface model.

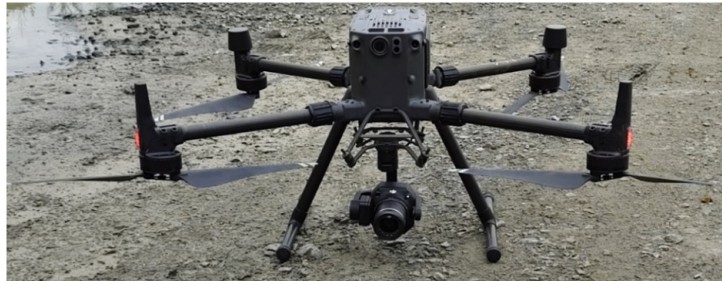

**Figure 2.** M300 RTK UAV with Zenmuse P1 sensor.

**Table 1.** The main technical parameters.

| Device | Category | Value |
| --- | --- | --- |
| M300 RTK UAV | Sizes | 430 × 420 × 430 mm (folded) |
| | Max. takeoff weight | 9 kg |
| | Max. flight altitude | 5000 m |
| | RTK position precision | 1 cm + 1 ppm (horizontal); 1.5 cm + 1 ppm (verticality) |
| Zenmuse P1 | Weight | 800 g |
| | Pixels | 45 million |
| | Precision | Plane: 3 cm, Elevation: 5 cm × GSD = 3 cm |

## 2.2. Reconstruction of Rock Mass Characteristics

After following the set route and collecting enough overlapping images of the slope surface, the SFM algorithm [24] was used to reconstruct the rock surface features. First, the feature points in the overlapping images are identified according to Scale Invariant Feature Transform (SIFT) [25], with at least three overlapping images, and as many images as possible should be taken for optimal feature point matching. Then, the feature points in the images, using Random Sample Consensus (RANSAC) algorithm [26], are connected according to the standard path to complete the point cloud sparse reconstruction. At the same time, the point cloud data collected by the UAV are calibrated to determine the position of the image control points in the point cloud, and the coordinates are calculated using the common WGS84 geodetic coordinate system. Finally, based on the Patch Based Multi-View Stereo (PMVS) algorithm, the sparse point cloud model is density enhanced to complete the dense reconstruction of the point cloud.

## 2.3. Discontinuities Extraction

The point cloud model of slope surface constructed based on UAV images is millions of 3D spatial coordinate points, and the spatial geometric information of the structural surface is stored in it. Geometric feature analysis of the point cloud data can identify the discontinuity information of the structural surface of the rock mass.

Based on the 3D point cloud model constructed by the UAV high-definition images, the identification of structural surfaces is completed by the method of multi-point fitting of structural surfaces [27]. The multi-point fitted structural surfaces will be explained in detail in the first paragraph of Section 3.2. The following three steps need to be completed to achieve the extraction of structural surface orientation: (1) estimation of point cloud normal vectors using *K*-nearest neighbor algorithm; (2) clustering using *K*-means algorithm to determine the set of structural surfaces; (3) eliminating noise points and solving for mean orientations.

(1)    Estimating point cloud normal vectors

The estimation algorithms on point cloud normal vectors can currently be classified into three types [28]: a method based on local surface fitting [29], a method based on Delaunay/Voronoi [30], and a method based on robust statistics [31]. For each of these three algorithms for estimating the point cloud normal vectors, it is difficult to compare them quantitatively in a comprehensive way [32]. Currently, *K*-nearest neighbor search is commonly used to find the neighboring points of the target point, and then the attribute features of the target point and the neighboring points are combined and analyzed. In order to perform the *K*-nearest neighbor search, firstly, each point in the point cloud is searched and analyzed according to the following principles: randomly selecting a target point, searching for $k$ ($k \geq 3$, three points that are not on a straight line can determine a plane) neighboring points of the target point, and then checking whether the target point and $k$ neighboring points are coplanar so as to obtain the optimal set of $k$ neighboring points of each point. After completing the *K*-nearest neighbor search, perform plane fitting to find out the normal vector of the target point. The least-squares fitted plane equation is

$$A\mathrm{x} + B\mathrm{y} + C\mathrm{z} + D = 0 \tag{1}$$

where *A*, *B*, *C*, and *D* are planar parameters, and any three cannot be zero at the same time; the normal vector of the plane is (*A*, *B*, *C*); *D* is a constant factor.

From the equation for the distance from the point to the plane (2), construct the equation for the sum of squares of deviations (3).

$$\mathrm{d} = \frac{\left| A\mathrm{x}_i + B\mathrm{y}_i + C\mathrm{z}_i + D \right|}{\sqrt[2]{A^2 + B^2 + C^2}} \tag{2}$$

$$S = \frac{1}{2} \sum_{i=1}^{k} (Ax_i + By_i + Cz_i + D)^2 \tag{3}$$

where $i$ is the number of $k$ neighboring point clouds, $i = 1, 2, \ldots, k$.

From the extreme value condition of Equations (4) and (5), the fitted values of the four planar parameters can be found and the normal vectors of the fitted plane can be obtained, and then the strike (the angle between the line of intersection of the rock face and the horizontal plane and the direction due north) and the dip angle can be calculated by the conversion equation [14,33].

$$\alpha = \frac{180}{\pi} \times \arctan \left| \frac{B}{A} \right| \tag{4}$$

$$\beta = \frac{180}{\pi} \times \arctan \left| \frac{\sqrt{A^2 + B^2}}{|C|} \right| \tag{5}$$

$$\alpha = \begin{cases} 0^\circ \sim 90^\circ \left( \text{or } 180^\circ \sim 270^\circ \right), & A \times B < 0 \\ 270^\circ \sim 360^\circ \left( \text{or } 90^\circ \sim 180^\circ \right), & A \times B > 0 \end{cases} \tag{6}$$

The calculation of the strike also needs to be based on Equations (4) and (6) to determine the range in which, when $A$ is 0, the strike is due east or due west, that is, 90° or 270°, and when $C$ is 0, the dip angle is 90°.

(2)　Clustering of structural surfaces

Some clustering algorithms on discontinuous sets are kernel density estimation algorithm [34], *K*-means algorithm [35], and fuzzy C-means algorithm [36]. The clustering *K*-means algorithm is a division-based unsupervised machine learning algorithm that requires manual input of the number of clusters *K*. Moreover, the selection of the initial cluster centers is random, and the clustering results are extremely sensitive to noise and outliers. However, the clustering of structural surface poles in the stereographic projection map is almost unaffected. Firstly, the structural surface is projected into the stereographic projection map, and the number and location of the clustering centers can be clearly judged according to the equal density map and the pole map of the structural surface; secondly, the range is artificially circled so that the clustering centers are moved to the area with the highest density intensity; finally, the outliers are eliminated according to the Fisher *K* value iterative calculation to find out the mean orientations.

The subdivision of the discontinuous sets is determined based on the similarity of the point cloud normal vectors. Jimenez-Rodriguez et al. [37] and Kong et al. [7] determine whether the structural surfaces are in the same group based on the distance similarity between point clouds. Jimenez, 2008 [38] and Xu et al., 2013 [39] utilized the pinch sine or sine square as a similarity metric. In this paper, we use the point cloud normal vectors angle threshold for determination, which is to compare the normal vectors of adjacent point clouds, and if their angle is less than or equal to the threshold $q$, they are classified as the same group of structural surfaces.

$$\arccos (t_i, t_j) \leq q \tag{7}$$

where $t_i$ and $t_j$ are the point cloud normal vectors; $i$ and $j$ are the number of the point cloud.

Concerning the value of $q$, there is no uniform regulation for the time being. Gao et al. [40] proposed distance thresholds of 0.005 and 0.12, which are converted to angles, and the angles between the point cloud normal vectors are 4° and 20°, respectively. Kong et al. [7] consider a lower limit of 0.005 and an upper limit of 0.12 for the distance threshold, where too large would result in too few groups and too small would result in too many groups. Likewise, when Peitao et al. [41] studied the point cloud model using the *K*-mean algorithm, the clustering effect was optimal for complex point cloud models with *K* of 7, an angle threshold of 17°, and a filtering noise factor of 5%, and the results were consistent with reality. Therefore, according to

the actual situation and the existing literature studies, the authors selected an angle threshold of 8 in the range of 4° and 20°, while setting *K* to 3. The clustering effect is consistent with reality. In Section 3.2, UAV-DP used *K*-mean clustering into three groups, which is the same as the manual measurement results, and the clustering effect is better. For the point cloud model of the multi-bench slope, the clustering effect is better by setting *K* as 6 and the angle threshold as 8, according to the polar map of the structure surfaces in the stereographic projection map.

(3)　Eliminating noise points and solving for mean orientations.

Since the poles of the same group of structural surfaces obey the Fisher distribution, the Fisher *K* value is used to iteratively calculate and eliminate the poles away from the mean directions until all boundary points are completely eliminated [42]. The effect of this iterative calculation can be seen in the stereographic projection map of the structure surfaces. In the isodensity map of the structured surfaces, iterative computation continuously eliminates boundary points, causing the center of clustering to keep moving toward the center of density (the center of density is the region with the darkest color).

Fisher's equation is as follows:

$$\mathrm{f}(\alpha) = \frac{K \sin(\alpha) e^{K \cos(\alpha)}}{e^K - e^{-K}} \tag{8}$$

where $\alpha (0 < \alpha < \pi/2)$ is the deviation of the structure surfaces orientations from the mean orientations; *K* is Fisher's constant, indicating the degree of dispersion, and the larger the value of *K*, the better the clustering effect.

When $K > 3$, *K* can be measured by the following probability distribution [43,44]:

$$K = \frac{M - 1}{M - |\boldsymbol{r}_M|} \tag{9}$$

where *M* represents the total number of structural surfaces in the same grouping, $\boldsymbol{r}_M$ represents the combined vector of all structural surfaces in the same grouping and the mean orientations, which can be derived from Equation (10) [45].

$$\delta_M = \cos^{-1} |Z_{rN}|; \ \theta_M = \tan^{-1} \left( \frac{x_r}{y_r} \right) \tag{10}$$

where $\delta_M$ is the dip angle; $\theta_M$ is the dip direction; $Z_{rN}$ is the component of the unit vector of the combined vector $\boldsymbol{r}_M$ in the Cartesian coordinate system on the Z-axis.

## 3. Engineering Application Study

The study area is located in the southwest part of the mine (Figure 3), in Taiyuan, Shanxi Province, China (Figure 4), and the slope body is about 900 m long and 176 m high. The typical geological section is shown in Figure 5. The lithology of the slope body is mainly mica–quartz schist, diagonal hornblende, and iron flash schist, of which mica-quartz schist is a laminated structure; diagonal hornblende is hard and broken and can form a block structure; iron flash schist is a laminated structure with low strength after weathering, and the joints and fissures are developed, which can form a body structure. From the site situation, this section of the slope is seriously cut by the structural surface, the stability of the slope is not optimistic, easy to be affected by excavation, blasting collapse, or rolling stone failure. In addition, there is a crushing station and tape routing below the slope, and the slope instability will seriously affect the normal operation of the mine and the safety of operators.

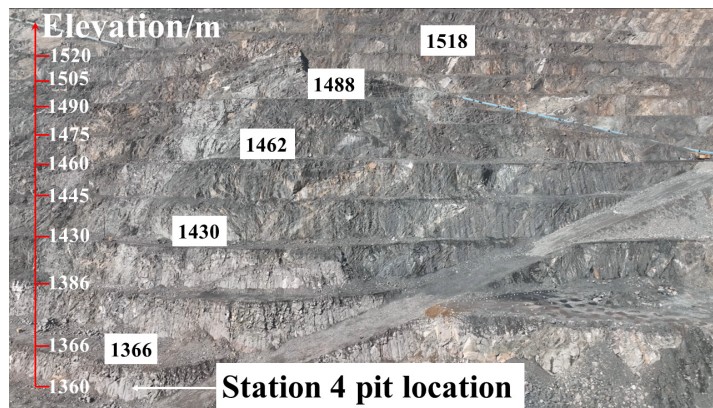

**Figure 3.** Southwest side slope of mine.

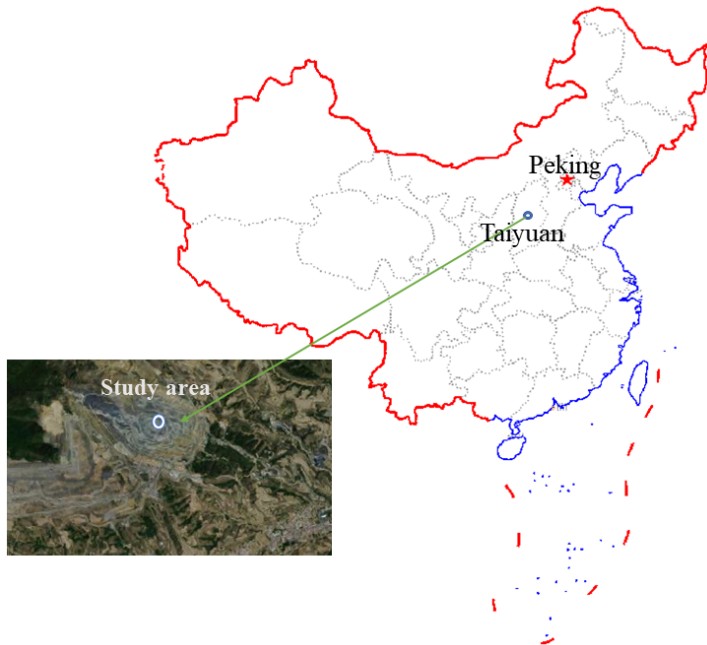

**Figure 4.** The study site is in Taiyuan, Shanxi Province, China.

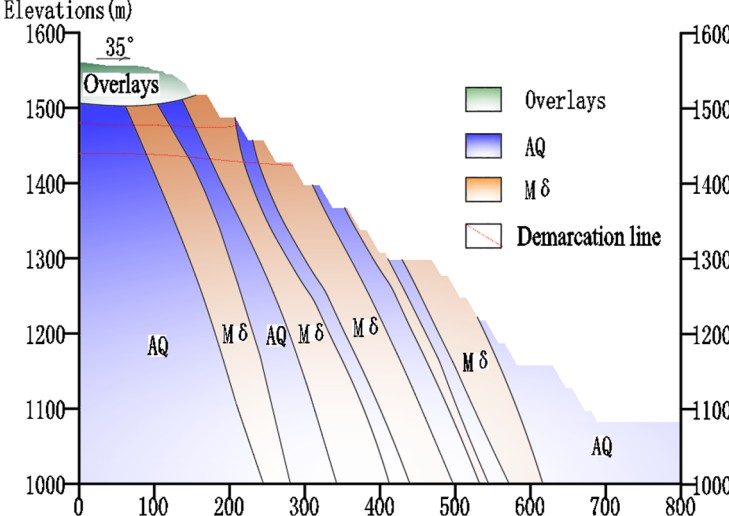

**Figure 5.** Southwest side slope of mine (AQ: Mica–quartz schist; Mδ: Plagioclase hornblende, Demarcation line: Rock weathering boundary).

Before stability evaluation of the slope, it is necessary to investigate the structural surface distribution of the rock mass. Since the measurement area was far from the foot of the slope and nearly vertical, 3D laser scanning and manual measurement could not find ideal observation points. Therefore, a UAV was used for photogrammetry of the slope. Meanwhile, the suitable area of platform 1366 m was selected for manual measurement and compared with the measurement results of the UAV. Based on the comparison results, the structural surface of the study area was determined and then clustered, which was then used for slope stability and failure model analysis.

### 3.1. Image Acquisition and Point Cloud Modeling

The accuracy of the 3D point cloud to construct a slope model is related to the UAV flight height, camera tilt angle, graphic overlap rate, number of control points, and the complexity of the terrain. Under the premise of ensuring flight safety and high resolution of images, the UAV was set within 120 m from the hillside. Moreover, the sensor lens was perpendicular to the hillside, and the heading overlap rate and lateral overlap rate were both 80%. In the geographic alignment of the 3D model, 13 GCPs were measured by using Huayi E300 RTK, and the specific points are shown in Figure 6a. The final 3D model of the slope surface in the study area obtained by the SFM method is shown in Figure 6b.

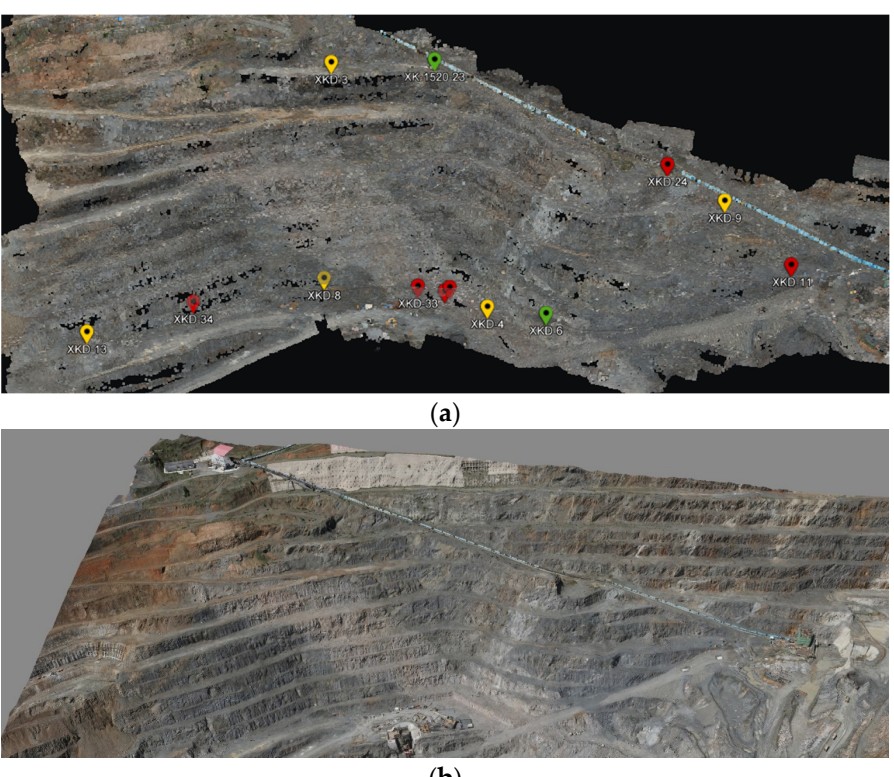

(**a**)

(**b**)

**Figure 6.** Digital surface modeling of the research area: (**a**) Rough modeling of slope and layout of GCPs; (**b**) Dense modeling of slope.

### 3.2. Discontinuity Extraction Precision Verification

To verify the precision of the UAV-DP technique, a wide, flat, and safe 1366 m-bench slope (Figure 7a) was selected for manual measurement and UAV-DP (Figure 7b), respectively. UAV-DP uses a multi-point fitting structural surface method to identify structural surfaces. In the surface model of the bench slope, the exposed rock structure faces are clearly visible. By clicking the mouse, the program will automatically pick up the representative point clouds with small undulation and wide distribution on the structure faces, which will automatically generate a plane, and the orientation information of the structure faces is the direction and dip angle of the plane where they are located. The colored patches

in Figure 7b serve to smooth the mesh for orientation extraction, and the different colors represent different structural surface groupings. The manual measurement method is the most widely used in geological surveys, using a tape measure to delineate a rectangular area of 20 m × 2 m and then using a geological compass to measure the orientations of the structural surfaces within the area. In the 1366 m-bench slope, 30 structural surfaces were collected by the manual survey method and 36 by the UAV-DP method. The structural surface information collected by both methods was analyzed by stereo-projection, and the results are shown in Figure 7c,d and Table 2.

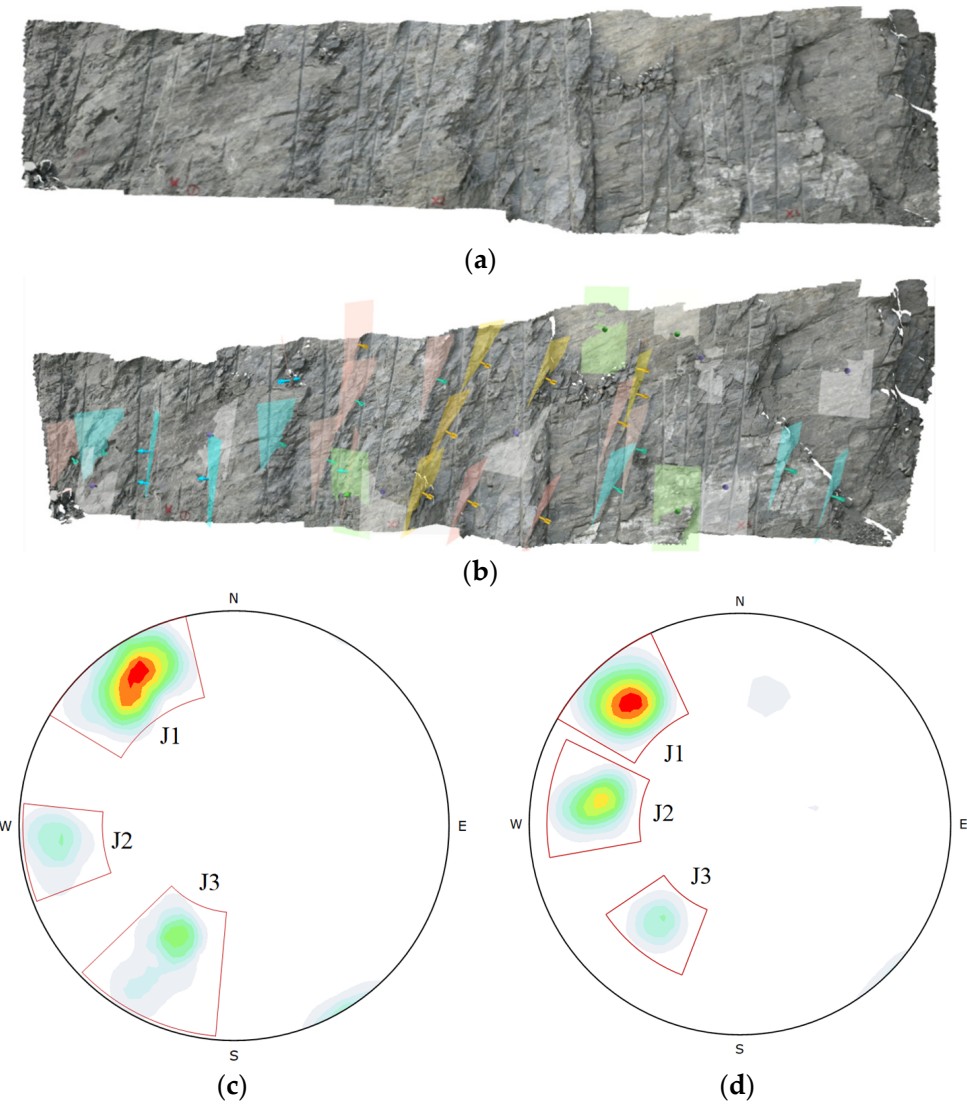

**Figure 7.** Identification and grouping of structural surfaces: (**a**) 1366 m bench slope of 3D model; (**b**) UAV photogrammetry to identify structural surfaces; (**c**) UAV-DP clustering group; (**d**) Manual measurement clustering group. The quadrilateral areas in Figure (**c**,**d**) are artificially circled grouping ranges.

**Table 2.** Comparison of structural surface clustering grouping results of two methods.

| Discontinuity Sets | UAV Dir/Dip (°) | Manual Dir/Dip (°) | Error Dir/Dip (°) |
|:---:|:---:|:---:|:---:|
| J1 | 146/78 | 142/76 | 4/2 |
| J2 | 96/79 | 99/75 | 3/4 |
| J3 | 35/60 | 41/62 | 6/2 |
| Mean error | | - | 4.33/2.67 |

Notes: Dir is direction, same as dip direction.

Through comparison, it was found that the maximum error of the tendency of the structure surface obtained by both was 6°, and the mean error was 4.33°; the maximum error of the inclination angle was 4°, and the mean error was 2.67°. Matsimbe, J. [46] concluded that the acceptable tolerance limits or errors between hand-drawn and remote data acquisition systems should be less than ±15°. It shows that the error is within the acceptable range, thus verifying the reliability of the UAV-DP. The subtle differences in the discontinuity direction may be due to inaccurate identification caused by the low density of the UAV point cloud, which makes it easy to overlook the fine structural surfaces. However, the fine structural surfaces do not affect the overall stability of the slope, so it is feasible to apply UAV-DP to the geological investigation of high-steep rock slopes.

### 3.3. Grouping of Structural Surfaces and Analysis of Failure Model

First, high-resolution images of each bench slope in the study area are obtained by UAV-DP. Then, the SFM algorithm was used to generate a dense point cloud model, and then the directions of all the rock structure faces were extracted. Finally, the dominant structural face group was obtained by clustering, and the results are shown in Figure 8 and Table 3.

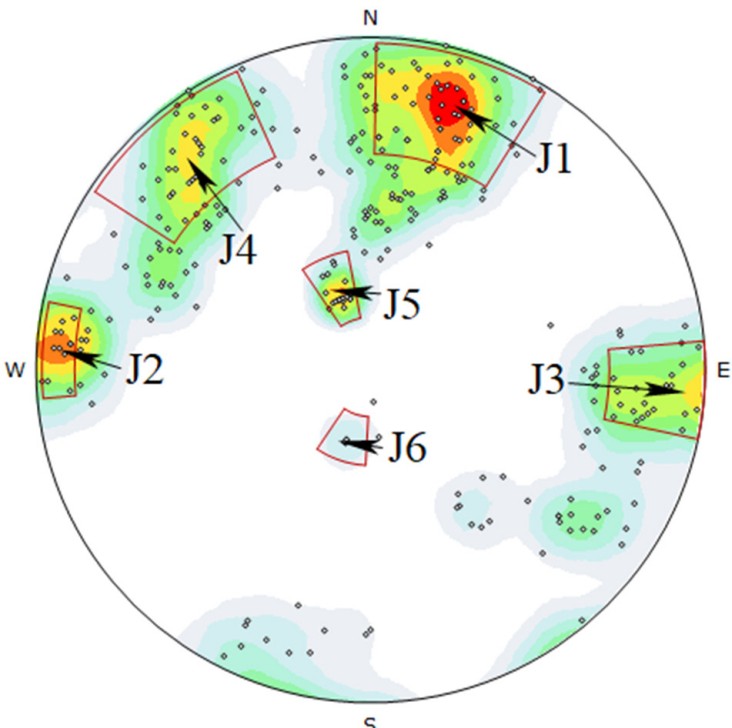

**Figure 8.** Polar and isodensity charts of the structural surface of the research area.

**Table 3.** Results of clustering group of structural surfaces acquired by UAV-DP.

| Discontinuity Sets | Dir (°) | Dip (°) | Proportions (%) |
|:---:|:---:|:---:|:---:|
| J1 | 191 | 71 | 38.3 |
| J2 | 92 | 81 | 7.3 |
| J3 | 271 | 75 | 10.7 |
| J4 | 132 | 86 | 28.3 |
| J5 | 163 | 28 | 7.3 |
| J6 | 19 | 25 | 2.0 |

Notes: Dir is direction, same as dip direction.

A total of 300 structural surfaces were identified from the point cloud model of the slope surface reconstructed from the UAV photographic images, and 282 structural surfaces were used for clustering grouping, accounting for 94%, among which, J2, J3, J5, and J6

structural surfaces were better clustered, and J1 and J4 structural surfaces were more discrete. The direction of the study area is between 34°~36°, and the slope angle is between 47°~49°. The stereo-projection analysis reveals that the potential failure model of the slope is mainly based on the flexural toppling of the J1 structure surface and the direct toppling of the J6 structure surface, as shown in Figure 9.

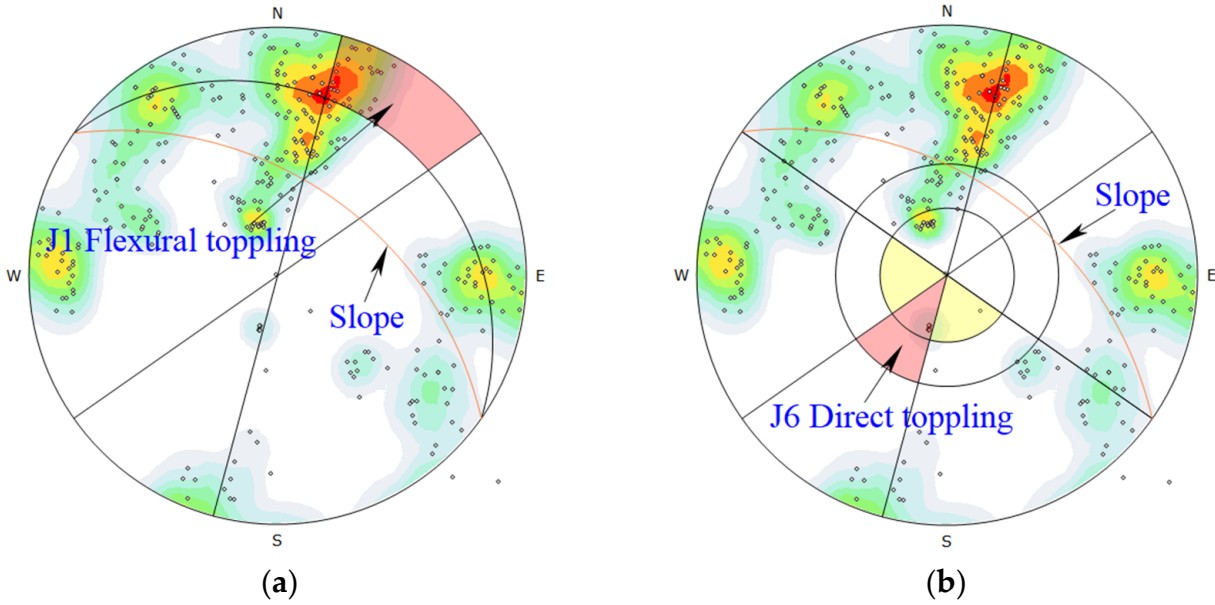

**Figure 9.** Potential failure types in the research area: (**a**) J1 structural surface flexural toppling; (**b**) J6 structural surface direct toppling.

### 3.4. Slope Stability and Deformation Characteristics Analysis

From the results of the above analysis, it can be seen that there are six groups of dominant structural faces in total developed in the rock mass of the slope in the study area, among which J1 and J6 structural faces are unfavorable to the stability of the slope. The rock body is cut by J1 and J6, and the integrity of the structure is destroyed, and it is easy to form toppling deformation failure under its own gravity and other external factors. The stability of the slope under the combination of J1 and J6 structural surfaces is analyzed by the strength reduction method of a discrete element, and the toppling deformation process of the slope is simulated to analyze its deformation and failure model. A typical geological profile was selected for conducting slope stability analysis (see Figure 4), which has a strike of 35° and slope angles of 47°~49°, and the apparent dip angles of J1 and J6 structural faces on the profile are 70° and 20°, respectively.

For calculation, the rock material adopts the Mohr–Coulomb elastoplastic model, and the structural surface adopts the Coulomb slip joint model; their physical and mechanical parameters are determined through comprehensive tests and previous research results, as shown in Table 4. The bottom boundary of the model is fixed, the left and right boundaries constrain the vertical displacement, and the upper boundary is free, considering only the effect of self-weight without considering other external loads.

**Table 4.** Parameters of rock mass and structural surfaces.

| Material | $\rho$ (kg/m$^3$) | $E$ (GPa) | $c$ (kPa) | $\varphi$ (°) | $k_n$ (GPa) | $k_s$ (GPa) |
|---|---|---|---|---|---|---|
| Rock mass | 2940 | 0.556 | 289 | 33.8 | - | - |
| J1 | - | - | 58 | 34.7 | 10 | 1 |
| J6 | - | - | 53 | 31.7 | 10 | 1 |

The calculation result of slope stability by the strength reduction method of a discrete element is shown in Figure 10; it can be seen that the safety factor of the slope under the action of self-weight is 1.12. According to GB51016-2014 Technical Specification for Non-Coal Surface Mine Slope Engineering, the safety reserve of the slope is not enough, and it may be destabilized under the action of external disturbances such as rainfall, blasting, and earthquake. The potential slip surface of the slope is basically straight at the foot of the slope, with an angle of about 3° in the direction normal to the anticline, and then extends upward in an arc, with the angle in the direction normal to the anticline gradually increasing to 20°, which is basically consistent with the previous research results [47].

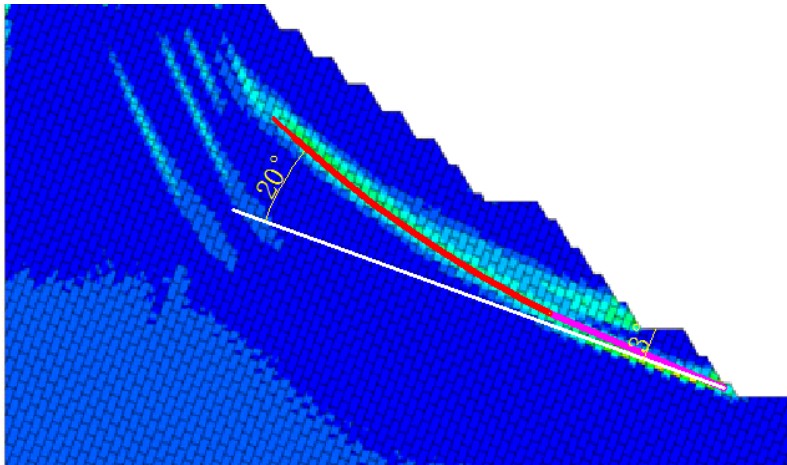

**Figure 10.** Results of the discrete element intensity discounting method (the Figure represents the strain increment diagram of the slope with a safety factor of 1.12).

After discounting the structural surface strength parameters, the slope dumping deformation failure process can be obtained, see Figure 11. It can be seen that the surface rock of the initial slope is deformed in the direction of the prograde under the action of self-weight, the internal displacement of the slope decreases along the level, and the overall toppling trend of the rock body is not obvious. After that, the rock body continued to deform and bend under the action of gravitational bending moment and gradually developed to the interior of the slope. The shear misalignment between the rock layers intensified, the structural surface separated, tension cracks appeared between the central plate and beam, and the surface layer collapsed and fell off the block. With the further development of rock bending deformation, the bending angle of the central plate beam further increases (about 13°), and the rest of the plate beam also gradually undergoes dumping deformation, and the tension cracks intensify, while the rock at the foot of the slope is extruded by the upper rock layer, and also gradually appears slight dumping deformation. On the whole, it seems that the deformation of the rock layer below the potential slip surface in this profile is small, while the anti-inclined rock layer above the potential slip surface in the middle is prone to overturning deformation and failure, which is the key part of slope instability failure and later reinforcement management.

The field investigation found that the thin-layered mica–quartz schist in the area of similar geological conditions in the vicinity was transformed into a dominant lamellar surface under the unloading condition, which caused the structural surface in the mica-quartz schist parallel to the surface to be highly developed. Under the long-term action of gravity, the shallow surface layer of rock gradually bends and deforms downward until the tensional fracture is destroyed, thus forming tensional fractures. This fissure continues to develop under the dissolution of rainwater and the action of gravity, gradually penetrates, and finally appears to collapse and fall off the block (Figure 12).

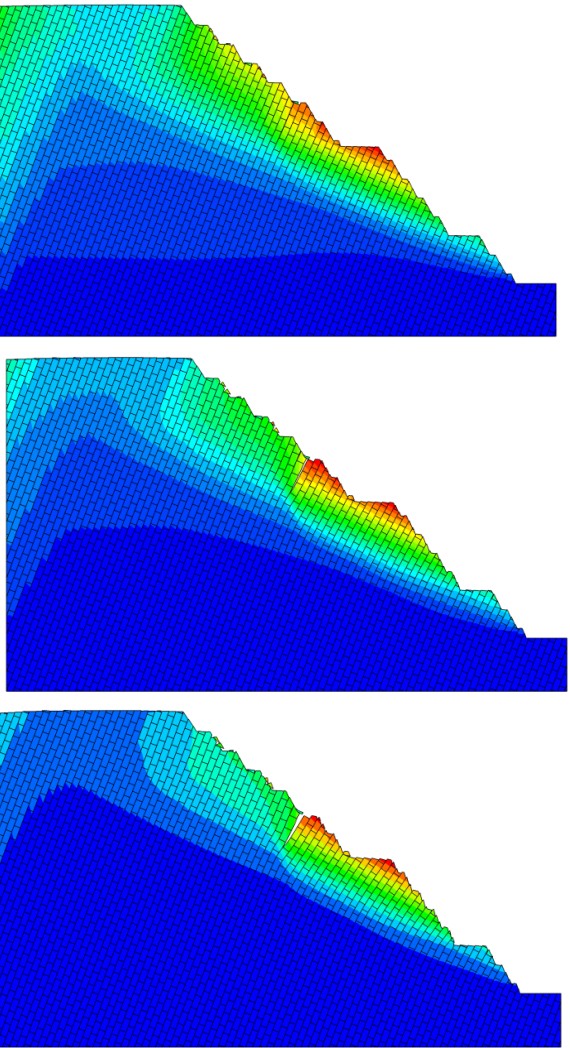

**Figure 11.** Slope toppling deformation failure process (the graph represents the total displacement change of the toppling process).

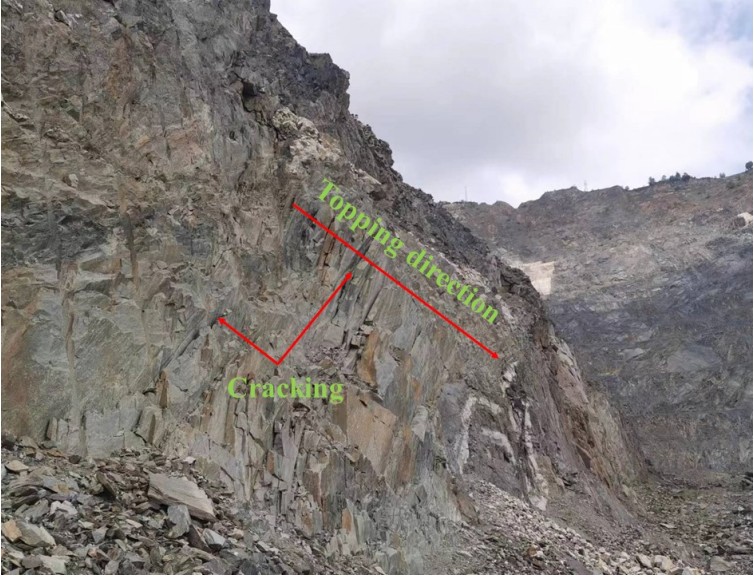

**Figure 12.** On-site tension cracking—Topping direction.

## 4. Conclusions

This paper describes the working method and data processing principle of UAV-DP technology in the geological investigation of high-steep rock slopes. The following conclusions are obtained when applying the method to the investigation of structural surfaces and stability evaluation of an iron ore mine:

(1) Compared with the traditional manual measurement method, the average directional error of the structure surface orientations obtained by UAV-DP is less than 5°, and the slight difference may be caused by insufficient point cloud density, which does not affect the engineering application, verify the feasibility of UAV-DP technology in the geological survey of high-steep rock slopes.

(2) There are six groups of dominant structural surfaces developed in the slope of the study area. The safety factor of the slope is 1.12, calculated by the distinct element-strength reduction method, which is not a high safety reserve and may be failed by toppling deformation under the cutting of J1 and J6 structural surfaces. This is similar to the failure model of the slope in the adjacent area observed in the field.

UAV-DP technology provides a reliable method for geological investigation and later stability evaluation of slopes, but it cannot completely replace the traditional manual measurement method at present because some information obtained by experienced geologists in field surveys cannot be identified by UAVs yet, such as water outflow point roughness and the filling of the structural surface. Therefore, the UAV-DP technology can make a breakthrough in these aspects in the future.

**Author Contributions:** Conceptualization, J.H., X.Z., C.W., H.W. (Hao Wang) and H.W. (Haibin Wang); methodology, J.H. and X.Z.; software, X.Z., C.W. and H.W. (Haibin Wang); validation, J.H. and X.Z.; formal analysis, J.H. and X.Z.; investigation, J.H., X.Z., C.W., H.W. (Hao Wang) and H.W. (Haibin Wang); resources, X.Z., H.W. (Hao Wang) and H.W. (Haibin Wang); data curation, J.H., X.Z. and C.W.; writing—original draft preparation, J.H. and X.Z.; writing—review and editing, X.Z., C.W., H.W. (Hao Wang) and H.W. (Haibin Wang). All authors have read and agreed to the published version of the manuscript.

**Funding:** This work was supported by the National Nature Science Foundation of China (Grant No. 42207222). The authors are solely responsible for its content.

**Institutional Review Board Statement:** Not applicable.

**Informed Consent Statement:** Not applicable.

**Data Availability Statement:** Not applicable.

**Acknowledgments:** The views, opinions, and recommendations expressed herein are solely those of the authors and do not necessarily reflect the views of the National Nature Science Foundation of China. Mentions of trade names, commercial products, or organizations does not imply endorsement by the authors nor the funding organization.

**Conflicts of Interest:** The authors declare no conflict of interest.

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
