# Peer review of "Application of UAV Digital Photogrammetry in Geological Investigation and Stability Evaluation of High-Steep Mine Rock Slope"

_drones, doi:10.3390/drones7030198_

Round 1
Reviewer 1 Report
See the attachement
Thanks and Good luck

Author Response
非常感谢您的评论,我已经以文字形式上传了具体的回复。

Reviewer 2 Report
This manuscript uses UAV photogrammetry technology to construct a fine 3D point cloud model of high and steep mine rock slopes for geological survey and stability evaluation, and it has a sound structure, fluent language and a certain degree of validation. But some details are missed, for example, several figures in the paper do not indicate the legend; the meaning of some symbols in section 2.3 is not explained, and no specific threshold is given, which may cause other researchers having difficulties in repeating the authors’ work; there are doubts about the clustering method in Section 3.3 and so on. Hope that the author will make appropriate revisions and answers, the details of which are as follows:
Line 61,64,147: Reference citation confusion.
Figure 1: What is the means and content of manual measurement? Can manually acquired data be used directly for discontinuous acquisition without preprocessing? It is recommended to add appropriate content to the manual collection of data.
Line 124 (Section2.3): Perhaps specific parameter values should be given in discontinuous extraction to give researchers a reference.
Equation 1:q represents the threshold of the angle, so whether the inverse trigonometric function arccos should be used? What i and j stand for and what their range of values is?
Equation 3:The meaning of n in the equation should be given.
Line 152: M represents the sample of what is in a structural surface group?
Figure 4: what does the symbol¥represent?
Figure 6 (b): What are the colored plaques in the picture?
Figure 6 (c), (d); Figure 7: The triangle symbol, cross symbol, and square symbol in the figure should give a legend.
Table 2: The unit of the error band in the first row of the table is %, but the actual calculated error unit in the table is °.
Line 211-212: Is the error reference range provided in Literature 28 universal for all rock slope studies without any restrictions? If so, perhaps relevant elements should be added to the paper, and if not, how should the appropriate margin of error be determined?
Figure 8: There is no legend for square symbols.
Table 3: K-means clustering is used here, but as we know, this clustering method has many limitations, why is this clustering method used in the experiment? Also, How are the k-values obtained in the experiment? Can you get better results using other clustering methods?
Author Response
非常感谢您的评论,我已经以文字形式上传了具体的回复。

Round 2
Reviewer 2 Report
In the revised draft, the author made changes to some questions and suggestions raised, but did not mark the modified parts, making the reviewer quite hard and time-consuming. Besides, there are still key concerns that are not addressed, making reviewer regards the method is not valid enough and cannot be repeated. Still too many potential detailed flaws, and recommend a thorough and careful revision. Authors should have corrected many flaws easily, but actually not. It is still below the publishing quality.
Reply to (d): cos(ti,tj) calculates the cosine value, not angle. And still not answered the meaning of i, j and the range of values. All formulas should have explanations for the letters and subscripts.
Reply to (h): In the reply, the authors gave explanations for the colored plaques in the figure, but did not add to the paper.
Reply to (I): The blue font in Figure 8 is not visible in the revised version, and perhaps a description of the square symbol should be added where appropriate.
Reply to (m): The reviewer does not agree. Actually, K-means is the one that is sensitive to outliers. And why using K-means is still not clear. And how to set K is still not explained in the manuscript. Reviewer may regard the method is not valid enough and cannot be repeated.
Author Response
Dear Editors and Reviewers:
We appreciate for your warm work earnestly. First of all, I apologize to the reviewers for the inconvenience caused by the first revision, which was not clearly marked. Secondly, thank the reviewers for their detailed comments on our manuscript entitled “Application of UAV Digital Photogrammetry in Geological Investigation and Stability Evaluation of High-Steep Mine Rock Slope” (ID: 2256786). Those comments are very valuable for improving the manuscript, and have important guiding significance for our researches. According to the reviewers’ suggestions, we have tried our best to improve the manuscript. The main modifications and the responds to the reviewer’s comments are as following:
Responds to the reviewer’s comments:
Reviewer #2
|
Number |
Comments |
Responses |
|
1 |
Reply to (d): cos(ti,tj) calculates the cosine value, not angle. And still not answered the meaning of i, j and the range of values. All formulas should have explanations for the letters and subscripts. |
Based on your comments, I changed it to an inverse trigonometric function. By reviewing the relevant literatures, I also find that it is more appropriate to change to inverse trigonometric functions. where and are the point cloud normal vectors; I and j are the number of the point cloud. Page 5 Line 188~204. |
|
2 |
Reply to (h): In the reply, the authors gave explanations for the colored plaques in the figure, but did not add to the paper. |
The explanation of the colored plaques is added to the manuscript. Page 8 Line 270~277. |
|
3 |
Reply to (I): The blue font in Figure 8 is not visible in the revised version, and perhaps a description of the square symbol should be added where appropriate. |
The font in Figure 8 has been turned up and arrows have been added. Page 10 In figures (c) and (d), the explanation of the quadrilateral region is supplemented in the manuscript. Page 9 Line 292~293. |
|
4 |
Reply to (m): The reviewer does not agree. Actually, K-means is the one that is sensitive to outliers. And why using K-means is still not clear. And how to set K is still not explained in the manuscript. Reviewer may regard the method is not valid enough and cannot be repeated. |
Moreover, the selection of the initial cluster centers is random, and the clustering results are extremely sensitive to noise and outliers. However, for the clustering of structural surfaces poles in the stereographic projection map is almost unaffected. Firstly, the structural surface is projected into the stereographic projection map, and the number and location of the clustering centers can be clearly judged according to the equal density map and the pole map of the structural surface; secondly, the range is artificially circled so that the clustering centers are moved to the area with the highest density intensity; finally, the outliers are eliminated according to the Fisher K value iterative calculation to find out the mean orientations. More detailed content is added in Page 5 line 168~179 of Section 2.3. |
Once again, thank you very much for your good suggestions. We look forward to your information about out revised manuscript.
Notes: The manuscript is marked for in-text revisions and is marked on the right side of the manuscript. Especially, the revisions of the manuscript focus on the abstract, section 2.3 and conclusion.
Yours sincerely,
Jianning Hao
